# Ginsenoside Rf Enhances Exercise Endurance by Stimulating Myoblast Differentiation and Mitochondrial Biogenesis in C2C12 Myotubes and ICR Mice

**DOI:** 10.3390/foods11121709

**Published:** 2022-06-10

**Authors:** Won-Chul Lim, Eun Ju Shin, Tae-Gyu Lim, Jae Woong Choi, Nho-Eul Song, Hee-Do Hong, Chang-Won Cho, Young Kyoung Rhee

**Affiliations:** 1Research Group of Traditional Food, Korea Food Research Institute, Wanju-gun 55365, Korea; godqhr1105@naver.com (W.-C.L.); sej296@naver.com (E.J.S.); choijw@kfri.re.kr (J.W.C.); nesong@kfri.re.kr (N.-E.S.); honghd@kfri.re.kr (H.-D.H.); 2Division of Food Science & Biotechnology, Sejong University, Seoul 05006, Korea; tglim@sejong.ac.kr

**Keywords:** exercise endurance, ginsenoside Rf, mitochondrial biogenesis, myogenesis

## Abstract

Ginsenoside Rf (G-Rf) is a saponin of the protopanaxatriol family and a bioactive component of Korean ginseng. Several ginsenosides are known to have a positive effect on exercise endurance, but there is not yet a report on that of G-Rf. Forced swimming tests were performed on G-Rf-treated mice to evaluate the effect of G-Rf on exercise endurance. Subsequently, the expression of markers related to myoblast differentiation and mitochondrial biogenesis in murine skeletal C2C12 myotubes and tibialis anterior muscle tissue was determined using Western blotting, quantitative real-time polymerase chain reaction, and immunofluorescence staining to elucidate the mechanism of action of G-Rf. The swimming duration of the experimental animal was increased by oral gavage administration of G-Rf. Moreover, G-Rf significantly upregulated the myoblast differentiation markers, mitochondrial biogenesis markers, and its upstream regulators. In particular, the mitochondrial biogenesis marker increased by G-Rf was decreased by each inhibitor of the upstream regulators. G-Rf enhances exercise endurance in mice, which may be mediated by myoblast differentiation and enhanced mitochondrial biogenesis through AMPK and p38 MAPK signaling pathways, suggesting that it increases energy production to satisfy additional needs of exercising muscle cells. Therefore, G-Rf is an active ingredient in Korean ginseng responsible for improving exercise performance.

## 1. Introduction

Skeletal muscle, which plays an important role in exercise and energy balance, has a remarkable regenerative ability in response to injury, illness, and aging [1]. Myogenesis is achieved by fusing myoblasts into multinuclear fibers (called myotubes) as a process of muscle tissue formation, particularly during embryonic development. During the differentiation process from myoblasts to myotubes, the expression of the myosin heavy chain (MHC), a representative marker, increases and is regulated by other muscle regulatory factors (MRFs) such as myoblast determination protein 1 (MyoD) and myogenin [2]. Mitochondria are essential eukaryotic organelles that play roles in oxidative phosphorylation and adequate adenosine triphosphate (ATP) production via various biological processes [3]. Mitochondrial biogenesis, which involves increasing the mitochondrial count and muscle endurance capacity for oxidative phosphorylation, is highly correlated with muscle function and energy production, resulting in enhanced exercise performance [4]. The 5′ adenosine monophosphate-activated protein kinase (AMPK) and p38 mitogen-activated protein kinase (p38 MAPK) signaling pathways are proposed to be involved in energy metabolism and regulation of glucose uptake, fatty acid oxidation, and mitochondrial biogenesis in skeletal muscle [5]. As a downstream factor of these pathways, peroxisome proliferator-activated receptor gamma coactivator 1 alpha (PGC-1α) activates transcription factors, such as nuclear respiratory factor 1 (NRF-1) and mitochondrial transcription factor A (TFAM), and subsequently triggers mitochondrial biogenesis and oxidative metabolism [6]. Thus, the upregulation of PGC-1α by the AMPK and p38 MAPK signaling pathways are associated with improved exercise capacity [5,7].

Korean ginseng (Panax ginseng Meyer), one of the most commonly used herbs worldwide for various pharmaceutical purposes [8], reportedly increases exercise performance, while black ginseng (processed ginseng obtained through repeated steaming and drying) increases muscle differentiation [9]. We have previously demonstrated that Korean red ginseng can enhance physical performance and induce myogenesis [10]. Ginseng contains various ginsenosides that are widely considered highly bioactive compounds [11], including ginsenosides Rg3, Rg1, Rb1, and Ro, which were shown to improve exercise performance and fatigue-associated factors in vivo [9,12,13]. Although ginsenoside Rf (G-Rf) is a representative ginsenoside uniquely present both in Korean ginseng and Korean red ginseng [14], there are no reports on its effect on exercise endurance.

In this study, we hypothesized that treatment with G-Rf, isolated from *P. ginseng*, may improve exercise endurance by regulating myoblast differentiation and mitochondrial biogenesis. Therefore, we evaluated the in vitro and in vivo effects of G-Rf on murine myoblasts in an effort to elucidate its mechanism of action.

## 2. Materials and Methods

### 2.1. Reagents

G-Rf was purchased from Fleton Natural Products Co., Ltd. (Chengdu, China) and dissolved in purified water for use in animal experiments. For cell experiments, G-Rf was dissolved in dimethyl sulfoxide (DMSO). The ATP assay kit was obtained from Abcam (Cambridge, MA, USA). Antibodies against NRF-1, phospho-AMPK, and phospho-p38 and secondary anti-rabbit or anti-mouse antibodies were purchased from Cell Signaling Technology (Beverly, MA, USA). MHC type 3 (MHCIII), AMPK, p38, and β-actin antibodies were purchased from Santa Cruz Biotechnology (Dallas, TX, USA). MyoD, myogenin, PGC-1α, and TFAM antibodies were obtained from Thermo Fisher Scientific (Waltham, MA, USA). The specific inhibitors including compound c, LY294002, and SB203580 were obtained from Cayman Chemicals (Ann Arbor, MI, USA).

### 2.2. Animals and Diet

ICR mice (5 wk-old males) were purchased from Dae Han Biolink (Eumseong-gun, Republic of Korea). All animal procedures were designed and conducted in accordance with the guidelines of the Institutional Animal Care and Use Committee (WJIACUC20190202-1-08) of Woojung Bio (Hwaseong, Republic of Korea). Animals were housed at a relative humidity of 50 ± 15% with a 12 h day/12 h night cycle and provided free access to food and pure water. Animals were acclimated for 1 wk prior to experimentation, randomly separated into three groups (*n* = 8), and then administered G-Rf dissolved in water (6 or 12 mg/kg body weight) daily for 4 wk by gavage (Figure 1A). The control group received an equivalent volume of water. Dietary intake and body weight were measured weekly.

### 2.3. Forced Swimming Endurance Test and Sample Collection

Mice were adapted to swimming through daily preliminary swimming sessions of 30–60 min for 1 wk. For the forced swimming test, a 5% body weight load was attached to the tail of each mouse, and the mice swam until exhaustion, i.e., when they could not rise to the surface within 7 s. After performing the forced swimming test, tibialis anterior muscle tissue was collected with the sacrifice and stored at −80 °C until further experiments.

### 2.4. Cell Culture, Differentiation, and Sample Treatment

Murine skeletal muscle C2C12 cells (American Type Culture Collection, Manassas, VA, USA) were cultured in Dulbecco’s Modified Eagle’s Medium containing 10% fetal bovine serum and incubated at 37 °C in a humidified CO_2_ incubator. Cells were cultured until they reached 90% confluence, and differentiation was initiated by exchanging the medium with Dulbecco’s Modified Eagle’s Medium containing 2% horse serum (Gibco, Grand Island, NE, USA; differentiation medium). G-Rf (10, 20, and 40 μM) dissolved in differentiation medium was added to the cells for 48 or 24 h on day 2 or 5 of differentiation, respectively (Figure 2A). Cells were treated with specific inhibitors of AMPK and p38 (compound C and SB203580, respectively; all 10 μM; Cayman Chemicals) for 24 h on day 5 of differentiation. DMSO-treated cells were used as a control and were compared with the experimental groups.

### 2.5. ATP Content Assay

On day 5 of cell differentiation, C2C12 cells were treated with G-Rf (0, 10, 20, or 40 μM diluted in differentiation medium) for 24 h and harvested to measure the ATP content (Figure 2A). Cells were lysed with ATP assay buffer and reacted with a mixture containing the ATP probe, converter, and developer for 30 min in the dark at room temperature. The ATP content was measured at 570 nm using a NanoQuant Infinite M200 Pro microplate reader (Tecan, Männedorf, Switzerland).

### 2.6. Cell Counting Kit 8 (CCK8) Assay

Cells were seeded in 96-well plates at a concentration of 5 × 10^3^ cells/well and cultured overnight. Then, cells were treated with G-Rf dissolved in DMSO at the indicated concentration and incubated for 24 or 48 h. The control group was treated with DMSO. After each incubation time, CCK8 reagent (EZ-3000, Dogen Bio, Seoul, Republic of Korea) was added to the wells for 1 h at a 1:10 ratio with complete medium and incubated at 37 °C. Cell viability was measured at 450 nm using a NanoQuant Infinite M200 Pro microplate reader.

### 2.7. Western Blotting

Cells were washed with cold phosphate-buffered saline (PBS) and lysed with cell lysis buffer (#9803, Cell Signaling Technology) for 10 min at 4 °C. After centrifugation at 12,000× *g* rpm for 10 min, the supernatant was collected, and the protein content was quantified using the Bradford assay. The quantified protein was mixed with sodium dodecyl sulfate sample buffer and heated at 95 °C for 5 min to prepare a loading sample. Proteins (10–30 μg/well) were loaded onto sodium dodecyl sulfate-polyacrylamide electrophoresis gels and developed at 220 V for 40 min for separation according to the molecular weight. The isolated proteins were transferred to polyvinylidene difluoride (PVDF) membranes for 6 min at 1.3 A, and the blotted membranes were incubated overnight at 4 °C with the indicated primary antibody. The membranes were washed 3× with Tris-buffered saline/Tween 20 buffer for 5 min, followed by incubation with the secondary antibody for 1 h at room temperature. Subsequently, cells were washed 3× with Tris-buffered saline/Tween 20 buffer for 5 min, exposed to enhanced chemiluminescence solution for 1 min, and imaged using a chemiluminescence reader. The protein level was calculated using ImageJ software (National Institutes of Health, Bethesda, MD, USA), and the relative level was obtained by comparison with the β-actin or total protein level.

### 2.8. Mitotracker and Immunofluorescence Staining

After cell differentiation, cells were stained with 50 nM MitoTracker (Cell Signaling Technology) for 45 min at 37 °C. Cells were then fixed with cold methanol for 10 min, followed by washing 3× with PBS for 5 min. Cells were permeabilized with 0.05% Triton X-100 solution, blocked with 1% bovine serum albumin solution for 30 min, and incubated overnight at 4 °C with the primary antibody. Cells were then washed 3× with PBS for 5 min and incubated with a goat anti-mouse immunoglobulin G, Flamma^®^ 488-labeled secondary antibody (Invitrogen, Waltham, MA, USA) for 1 h in the dark. Nuclei were stained with 4′,6-diamidino-2-phenylindole and observed using an Eclipse Ti-U fluorescent microscope (Nikon, Tokyo, Japan). The fusion index was determined by calculating the ratio of nuclei present in MHC-positive cells with 2 or more nuclei among the total nuclei in the randomly selected region.

### 2.9. Quantitative Real-Time Polymerase Chain Reaction (qRT-PCR)

The mRNA level of each factor in C2C12 cells was measured using qRT-PCR. For RNA extraction, easy-BLUE Total RNA Extraction reagent (Intron Biotechnology, Seongnam, Republic of Korea) was used according to the manufacturer’s instructions. Briefly, cells differentiated for 4 d in a 6-well plate were washed with cold PBS, mixed with 1 mL easy-BLUE Total RNA Extraction reagent, vortexed with 200 μL chloroform for 10 s, and centrifuged at 13,000× *g* rpm for 10 min. Then, 400 μL supernatant was mixed with an equal amount of isopropyl alcohol and incubated at room temperature for 10 min. After centrifuging the mixture at 13,000× *g* rpm for 5 min, the supernatant was removed and washed with 70% ethanol. After centrifugation at 10,000× *g* rpm for 5 min, the RNA pellet was dissolved in RNase-free water and quantified using a Nanodrop 2000 spectrophotometer (Thermo Scientific, Wilmington, NC, USA). ReverTra Ace qPCR RT Master Mix (Toyobo, Osaka, Japan) was used for cDNA synthesis, following these steps: RNA denaturation for 5 min at 65 °C, cDNA synthesis for 5 min at 50 °C, followed by enzyme inactivation for 5 min at 98 °C. The synthesized cDNA was subjected to real-time PCR using specific primers and SYBR Green Real-Time PCR Master Mix (Toyobo), under the following thermal conditions: preheating at 95 °C for 2 min, and 39 cycles of heating at 95 °C for 15 s, annealing at 60 °C for 20 s, and extension at 72 °C for 30 s. The specific primer sequences are listed in Table 1. The mRNA level of each factor was calculated using the 2^−ΔΔCT^ method.

### 2.10. Statistical Analysis

All experiments were performed at least in triplicate, and the results are presented as the mean ± standard deviation (*n* = 8). Statistical significance was calculated using the two-way ANOVA followed by Dunnett’s post hoc test, with *p*-value < 0.05 considered statistically significant.

## 3. Results

### 3.1. G-Rf Enhances Forced Exercise Endurance of ICR Mice

To measure whether G-Rf affects the exercise endurance of mice, the forced swimming test was conducted after G-Rf administration for 4 wk (Figure 1A). Dietary intake and body weight did not differ significantly between groups at 4 wk; average body weight of control, G-Rf-L, and G-Rf-H-treated groups after 4 wk were represented as 36.8, 36.6, and 36.6 g, respectively (Figure 1B,C). The swimming duration until exhaustion increased in the G-Rf-treated groups in a dose-dependent manner compared to the control group. In particular, mice treated with 12 mg/kg G-Rf (G-Rf-H group) showed a 1.23-fold longer swimming duration than mice in the control group (*p* < 0.05) (Figure 1D).

### 3.2. G-Rf Stimulates Muscular ATP Production of C2C12 Myotubes

First of all, CCK8 assay was performed to evaluate whether G-Rf affects the proliferation of C2C12 myoblasts (Figure 2B). Treatment with G-Rf did not affect C2C12 cell proliferation at concentrations of 0, 10, 20, and 40 μM G-Rf. Therefore, these G-Rf concentrations were used in cell experiments in which C2C12 myotubes were treated with G-Rf for 24 h, and intracellular ATP concentration was measured. G-Rf increased ATP production in C2C12 myotubes in a concentration-dependent manner (Figure 2C), confirming the effect of G-Rf on ATP production in vitro.

### 3.3. G-Rf Increases Differentiation of C2C12 Myoblasts

Western blotting analysis was performed to determine whether G-Rf treatment affects the expression of proteins associated with myogenesis. Expression of MHCIII, a differentiation marker in C2C12 myotubes, increased in a concentration-dependent manner upon G-Rf treatment (Figure 3A). Likewise, G-Rf treatment significantly upregulated the expression of MyoD and myogenin, which are known to be involved in the process of myoblast differentiation. The increase in muscle cell MHCIII expression triggered by G-Rf was confirmed by immunofluorescence staining (Figure 3B). G-Rf treatment significantly increased the fusion index of C2C12 myotubes, especially in cells treated with 40 μM G-Rf, exhibiting an approximately 10-fold increase compared to untreated cells (Figure 3C). The higher fusion index was consistent with the observed increase in MHCIII protein levels triggered by G-Rf, suggesting that G-Rf may increase muscle mass and improve exercise performance through structural differentiation of skeletal muscle.

### 3.4. G-Rf Stimulates Mitochondrial Biogenesis in C2C12 Myotubes

Mitochondria of C2C12 cells were evaluated using MitoTracker staining to confirm whether the enhanced endurance demonstrated by G-Rf-treated mice in the forced swimming tests was caused by mitochondrial biogenesis. The number of MitoTracker-stained mitochondria increased in a concentration-dependent manner after G-Rf treatment (Figure 3B). Moreover, protein and mRNA levels of related transcription factors were evaluated by Western blotting and qRT-PCR analysis, respectively. G-Rf treatment dose-dependently upregulated protein levels of PGC-1α, NRF-1, and TFAM compared to the untreated control (Figure 4A). Based on these results, the levels of mitochondrial DNA (mtDNA), a downstream factor of the mitochondrial biogenesis pathway, along with those of PGC-1α, NRF-1, and TFAM were evaluated by qRT-PCR analysis in the groups treated with 20 and 40 μM G-Rf. Concurring with the Western blotting results, G-Rf treatment significantly increased the mRNA levels of each factor (Figure 4B). As expected, mtDNA levels were enhanced by G-Rf treatment in a dose-dependent manner (Figure 4C). These results indicate that G-Rf treatment enhances the mitochondrial biogenesis of C2C12 myotubes.

### 3.5. G-Rf Improves Exercise Endurance by Activating the AMPK and p38 MAPK Signaling Pathways

Exercise stimulates energy production by regulating PGC-1α through activation of several signaling pathways, including those of AMPK and p38 MAPK [7]. Western blotting was performed to evaluate whether G-Rf affects these pathways in C2C12 cells. G-Rf stimulated phosphorylation of AMPK and p38 in a dose-dependent manner (Figure 5A). TFAM, regulated by PGC-1α, is involved in energy production by enhancing DNA transcription and mitochondrial function [15]. To determine whether G-Rf regulates TFAM through activation of these signaling pathways, cells were co-treated with 10 μM AMPK- and p38-specific inhibitors and 40 μM G-Rf. The increased TFAM expression induced by G-Rf treatment was impaired by each specific inhibitor (Figure 5B). These results imply that G-Rf activates the mitochondrial biogenic pathway by phosphorylating AMPK and p38. Next, to verify the stimulating activity of G-Rf on muscular differentiation and mitochondrial biogenesis in vivo, the expression of each factor was investigated in the tibialis anterior muscle tissue of experimental mice. G-Rf administration upregulated the expression of the differentiation markers MHCIII, MyoD, and myogenin, as well as mitochondrial biogenesis-related markers PGC-1α, NRF-1, and TFAM. The high-dose G-Rf-treated group, in particular, exhibited significant upregulation of these factors compared to the control group. In addition, G-Rf treatment increased the phosphorylation of AMPK and p38 in the tibialis anterior muscle tissue of experimental mice (Figure 5C).

## 4. Discussion

The forced swimming test is a commonly used method to evaluate the effect of active substances on the endurance or fatigue of experimental animals. In this study, the forced swimming duration of mice treated with G-Rf (6 and 12 mg/kg body weight) was significantly increased compared to the control group. In many studies dealing with ginsenosides, doses in the range of 5–20 mg/kg were applied to animal experiments [16,17]. In fact, according to the standards of the Korea Food and Drug Administration, the recommended daily intake of ginsenoside in red ginseng is 3–80 mg/day [18]. The 6 or 12 mg/kg used in this study are 28.8 or 57.6 mg/day when the conversion factor for body surface area is multiplied by 0.08 and 60 kg of adult body weight, which is similar to the recommended daily intake range [19]. Therefore, based on these facts, an appropriate dose of ginsenoside Rf was determined. Muscle contraction and sustained exercise require ATP production to satisfy the ATP demand generated by these processes [20]. Interestingly, G-Rf administration increased the ATP content in the muscle tissue of experimental mice compared to that of control mice. In particular, ATP production significantly increased in the high-dose G-Rf-treated group, which concurred with the extended swimming time. Therefore, to elucidate the mechanism of this in vitro, expression of factors necessary for ATP production in C2C12 muscle cells was confirmed. In most studies using C2C12 cells, samples were treated on the fifth day of differentiation and the amount of ATP was measured on the sixth day [21,22].

In order to generate ATP in C2C12 myotubes, successful differentiation from myoblasts to myotubes is required. In this study, G-Rf was presumed to accelerate muscle cell differentiation by upregulating the expression of markers involved in early stages of differentiation, including MHCIII, MyoD, and myogenin. MyoD, an MRF that directs skeletal muscle development, is known to be expressed at the onset of differentiation, which terminates myoblast proliferation and includes the expression of myogenin and MHC [2]. Myogenin, along with MyoD, is a member of the MRF family required for proper differentiation of most myogenic precursor cells. Some studies have shown that myogenin deficiency causes severe skeletal muscle disorders [23]. MHC is a major component of muscle fibers, and MHCIII is known to be part of the myosin protein complex, which is important for early muscle development [24]. Therefore, MHC is proposed as a representative marker for differentiation of C2C12 cells in numerous studies [25]. Myotubes, the differentiated form of myoblasts, are structures consisting of adjacent muscle cells fused to each other and containing several nuclei. Different studies have suggested that the degree of cell differentiation can be assessed through the cell fusion index [10,26]. As expected, G-Rf treatment significantly increased the fusion index, concurring with upregulated MHCIII expression.

Increased mitochondrial biogenesis, which enhances the maximum capacity of skeletal muscle to produce ATP by oxidative phosphorylation, is a hallmark of the response to exercise [27]. PGC-1α is a transcriptional coactivator that regulates genes involved in energy metabolism and a major regulator of mitochondrial biogenesis. This protein regulates the activity of transcription factors, such as the cyclic adenosine monophosphate response element-binding protein (CREB) and NRF, through interaction with nuclear receptor peroxisome proliferator-activated receptor gamma (PPAR-γ). This primarily occurs in slow-twitch muscle fibers as a link between physiological external stimuli and regulation of mitochondrial biogenesis. Endurance exercise is known to activate and upregulate PGC-1α in human skeletal muscle. Based on in vitro experiments, NRF-1/2 binds to the TFAM promoter and upregulates the transcription of several nuclear-encoded respiratory genes. TFAM then binds to the promoter of mtDNA to initiate its transcription and replication [27]. Located in the mitochondria, mtDNA converts chemical energy into ATP to make energy available to the cell. In this study, G-Rf treatment increased ATP production in C2C12 skeletal myotubes, which presumably improved energy production by further upregulating PGC-1α, NRF-1, TFAM, and mtDNA levels.

Numerous researchers have suggested that PGC-1α expression is regulated in response to the stimulation of several upstream regulatory pathways such as the AMPK and p38 MAPK pathways [5]. AMPK, a crucial sensor of the cellular energy state, is activated when the adenosine monophosphate/ATP ratio increases, triggering the catabolic pathways required to increase ATP levels in cells. Therefore, AMPK in the muscle is activated during exercise, which improves mitochondrial biogenesis, one of the catabolic pathways. In fact, 5-aminoimidazole-4-carboxamide-1-β-d-ribofuranoside (AICAR), an AMPK activator, is known to increase PGC-1α transcription in muscle cells [28]. Another mechanism that regulates exercise-induced PGC-1α expression in muscle involves p38 MAPK activation. This kinase increases PGC-1α expression through activation of myocyte enhancer factor 2 (MEF2) and activating transcription factor 2 (ATF2). In addition, physical exercise regulates a number of metabolic and transcriptional events in skeletal muscles. In the current study, G-Rf treatment induced significant upregulation of PGC-1α, a major regulator of energy production in skeletal muscle, through phosphorylation of AMPK and p38 MAPK. These results suggest that the increased swimming endurance demonstrated by G-Rf-treated mice was enabled by satisfying the additional energy requirements of muscle cells during exercise, which in turn occurred through the activation of the AMPK and p38 MAPK signaling pathways and increased mitochondrial biogenesis.

Numerous studies have investigated the various effects of Korean ginseng, such as improving immunity and exercise capacity, and have proven its benefits to health; however, endurance enhancement by G-Rf was not yet reported. According to previous studies, the endurance improvement effect of red ginseng containing G-Rf supports the beneficial effect of G-Rf on endurance revealed in this study. Although research on a single ginsenoside is important, investigating the synergistic effects of specific combinations of individual ginsenosides on functionality remains necessary. In addition, although this study confirmed the beneficial effect of G-Rf on endurance in vitro and in vivo, it is necessary to verify the effect through human trials and suggest an administration strategy.

## 5. Conclusions

In summary, G-Rf was selected in an effort to discover the active ingredient responsible for the endurance-enhancing effect of Korean ginseng demonstrated in previous studies. The effect of G-Rf treatment was examined on the swimming endurance of mice, murine skeletal muscle C2C12 cells, and mouse tibialis anterior muscle tissue. Our findings suggest that G-Rf treatment promotes muscle cell differentiation and mitochondrial biogenesis by activating the AMPK and p38 MAPK signaling pathways, thereby increasing the production of energy required for exercise and consequently increasing exercise endurance. Therefore, G-Rf shows potential as an effective natural supplement for improving exercise endurance through positive regulation of muscle cell differentiation and mitochondrial biogenesis. The results of this study broaden the applications of Korean ginseng and provide a novel functional ginsenoside for future research to improve exercise performance and develop functional foods, thereby securing the global competitiveness of this product.

## Figures and Tables

**Figure 1 foods-11-01709-f001:**
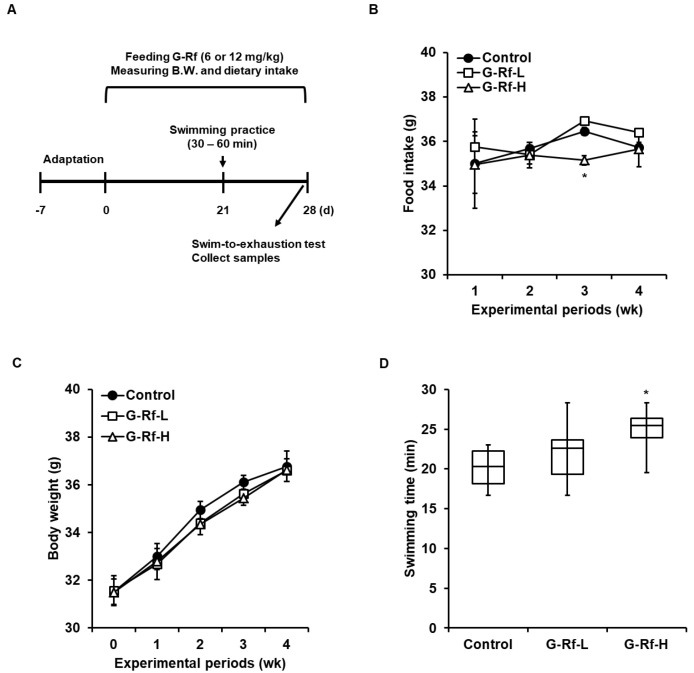
G−Rf enhances forced exercise endurance in ICR mice. (**A**) Schematic of animal experiments related to the forced swimming test. Mice were orally administered G−Rf dissolved in water (6 or 12 mg/kg) daily for 4 wk. The control group was administered the equivalent volume of water. (**B**) Food intake and (**C**) body weight were measured every week for 4 wk. (**D**) Swimming duration was recorded until mice did not rise above the water within 7 s due to exhaustion. Data are expressed as the mean ± standard deviation (*n* = 8). * *p* < 0.05 compared to the control group.

**Figure 2 foods-11-01709-f002:**
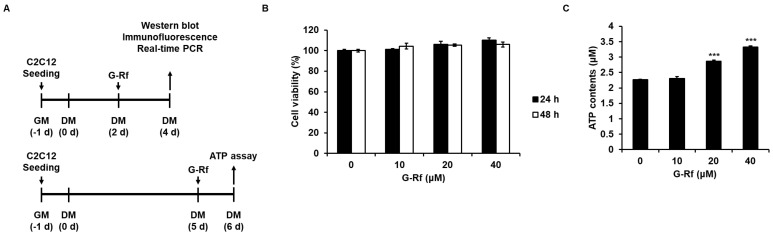
G−Rf enhances ATP levels in murine skeletal C2C12 myotubes. (**A**) Schematic of C2C12 myoblast differentiation experiments. (**B**) C2C12 cells were treated with G−Rf at the indicated concentrations for 24 or 48 h, and then cell viability was investigated using the CCK8 assay. (**C**) C2C12 cells were treated with G−Rf at the indicated concentrations for 24 h and harvested to measure intracellular ATP levels using the colorimetric ATP content assay. Data are expressed as the mean ± standard deviation (*n* = 3). *** *p* < 0.005 compared to untreated myotubes.

**Figure 3 foods-11-01709-f003:**
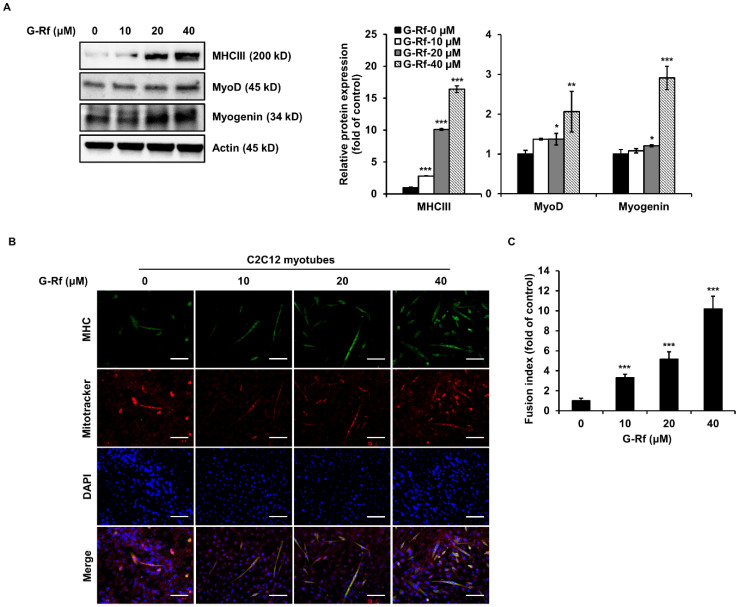
G-Rf increases myogenic differentiation and mitochondrial biogenesis in C2C12 myoblasts. On the second day of differentiation, C2C12 cells were treated with 10, 20, or 40 μM G-Rf for 48 h, and the expression of the myogenic differentiation markers MHCIII, MyoD, and myogenin was determined by (**A**) Western blotting and (**B**) immunofluorescence analysis (scale bar: 100 μm). (**C**) Fusion index of myotubes. Data are expressed as the mean ± standard deviation (*n* = 3). * *p* < 0.05, ** *p* < 0.01, *** *p* < 0.005 compared to untreated myotubes.

**Figure 4 foods-11-01709-f004:**
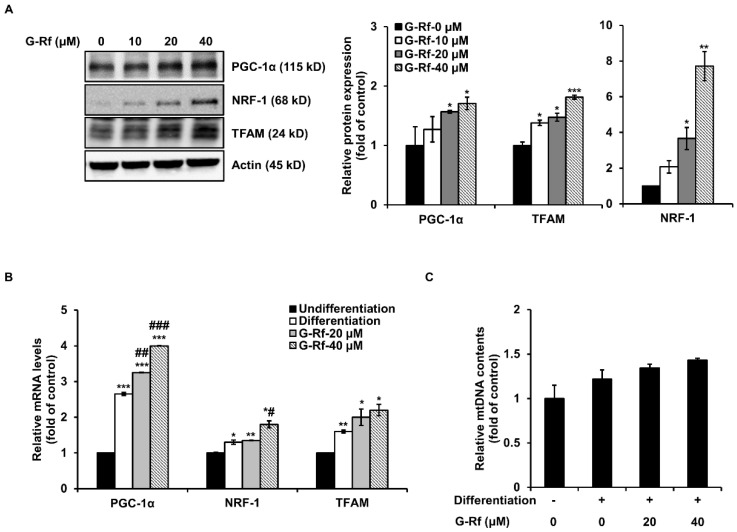
G-Rf upregulates the levels of markers associated with mitochondrial biogenesis in C2C12 myotubes. (**A**) Protein and (**B**) mRNA levels of PGC-1α, NRF-1, and TFAM were measured using Western blotting and qRT-PCR, respectively. (**C**) Total mtDNA was investigated using qRT-PCR. Data are expressed as the mean ± standard deviation (*n* = 3). * *p* < 0.05, ** *p* < 0.01, *** *p* < 0.005 compared to untreated myoblasts. ^#^
*p* < 0.05, ^##^
*p* < 0.01, ^###^
*p* < 0.005 compared to untreated myotubes.

**Figure 5 foods-11-01709-f005:**
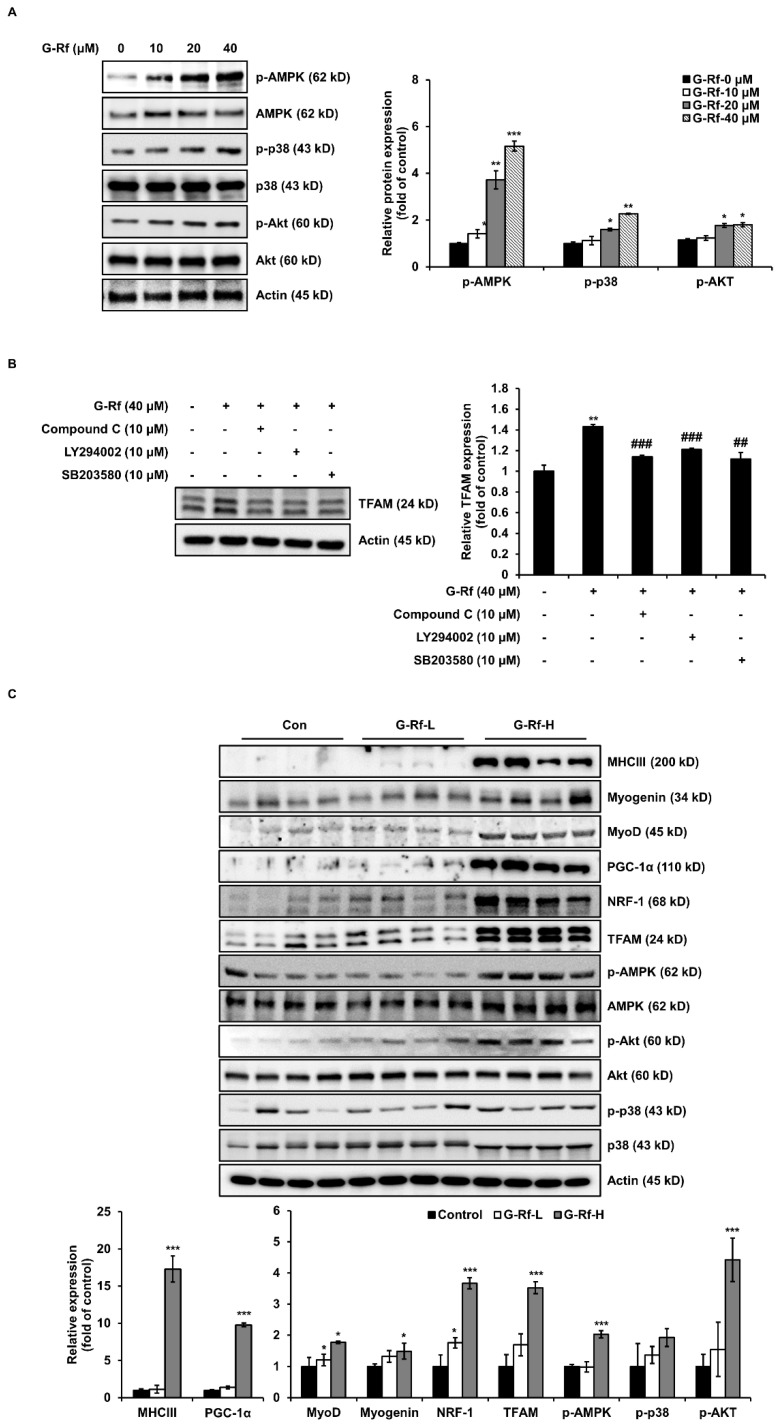
G-Rf enhances exercise endurance by activating phosphorylation of AMPK and p38. (**A**,**B**) Cells were lysed to determine the levels of p-AMPK, AMPK, p-p38, p38, and TFAM using Western blotting. (**C**) The expression of biomarkers associated with myogenic differentiation or mitochondrial biogenesis in tibialis anterior muscle tissue were determined using Western blotting. Data are expressed as the mean ± standard deviation (*n* = 3). * *p* < 0.05, ** *p* < 0.01, *** *p* < 0.005 compared to the untreated group. ^##^
*p* < 0.01, ^###^
*p* < 0.005 compared to G−Rf treated group.

**Table 1 foods-11-01709-t001:** List of primer sequence.

Gene	Direction	Sequence (5′ to 3′)
PGC-1α	Forward	TATGGAGTGACATAGAGTGTGCT
	Reverse	CCACTTCAATCCACCCAGAAAG
NRF-1	Forward	CCATCTATCCGAAGAGACAGC
	Reverse	GGGTGAGATGGCAGAGTACAATC
TFAM	Forward	GGAATGTGGAGCGTGCTAAAA
	Reverse	GCTGGAAAAACACTTCGGAATA
GAPDH	Forward	CATGGCCTTCCGTGTTCCTAC
	Reverse	TCAGTGGGCCCTCAGATGC
mtDNA	Forward	CGTTAGGTCAAGGTGTAGCC
	Reverse	CCAGACACACTTTCCAGTATG
β-Actin	Forward	GATTACTGCTCTGGCTCCTAGC
	Reverse	GATTACTGCTCTGGCTCCTAGC

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
