# Peer review of "Ginsenoside Rf Enhances Exercise Endurance by Stimulating Myoblast Differentiation and Mitochondrial Biogenesis in C2C12 Myotubes and ICR Mice"

_foods, 2022, doi:10.3390/foods11121709_

Round 1
Reviewer 1 Report
The purpose of this study was two-fold: 1) to investigate if Ginsenoside Rf influence endurance swimming capacity in mice. 2) identify potential molecular mediators of mitochondrial and muscle differentiation in C2C12 cells with Ginsenoside Rf administration. Overall, I like the focus of the paper and I think authors have done a nice job here. Below are my comments.
Check English and grammar throughout.
Throughout the manuscript, authors should temper their language about drawing conclusions from cell culture experiments and translating them to the whole animal model. For example, authors say “G-Rf enhances exercise endurance via myoblast differentiation and mitochondrial biogenesis by activating the AMPK and p38 MAPK signaling pathways…” in the abstract. However, a myoblast cell line and mature in vivo skeletal have many similarities but are not equal. Authors should use less matter-of-fact language such as “G-Rf enhances exercise endurance in mice which may be mediated by myoblast differentiation and enhanced mitochondrial biogenesis through AMPK and p38 MAPK signaling.”
Line 41: change to “muscle endurance capacity”
Authors should add a paragraph in the introduction giving background to previous work on Korean ginseng and exercise. The rest of the introduction is written nicely.
Line 75: which specific inhibitors? Phosphatase inhibitors? Can authors include clarification?
Line 84: Administered how? Gavage or was water consumption used to calculate?
Authors should add the molecular weight next to their blots. It can simply be noted to the side of the blot in the figures.
I want to commend the authors for the blots in figure 5. Should sample lanes in quadruplicate strengthens your findings and shows transparency.
Authors need to list the limitations of the study. For example, the translation of this to humans is unknown. Also, the most effective dosing strategies remain to be discovered so these findings may not translate to all dosages and regimens.
Author Response
Comment 1: Throughout the manuscript, authors should temper their language about drawing conclusions from cell culture experiments and translating them to the whole animal model. For example, authors say “G-Rf enhances exercise endurance via myoblast differentiation and mitochondrial biogenesis by activating the AMPK and p38 MAPK signaling pathways…” in the abstract. However, a myoblast cell line and mature in vivo skeletal have many similarities but are not equal. Authors should use less matter-of-fact language such as “G-Rf enhances exercise endurance in mice which may be mediated by myoblast differentiation and enhanced mitochondrial biogenesis through AMPK and p38 MAPK signaling.”
Answer 1: I appreciate your comment. As your comment, in lines 25-27 of the abstract, the previous language about concluding was changed to minor matter-of-fact language.
Comment 2: Line 41: change to “muscle endurance capacity”
Answer 2: As your comment, “..cellular capacity..” was changed to “..muscle endurance capacity..”. (line 57 in the revised manuscript)
Comment 3: Authors should add a paragraph in the introduction giving background to previous work on Korean ginseng and exercise. The rest of the introduction is written nicely.
Answer 3: As your comment, the correlation between Korean red ginseng and exercise was described in lines 71-72 of the introduction section. According to reference 10, Korean red ginseng, including ginsenoside Rf enhanced exercise endurance.
Comment 4: Line 75: which specific inhibitors? Phosphatase inhibitors? Can authors include clarification?
Answer 4: As your comment, “The specific inhibitors were obtained…” was changed to “The specific inhibitors including compound c, LY294002, and SB203580 were obtained…”. (line 92 in the revised manuscript)
Comment 5: Line 84: Administered how? Gavage or was water consumption used to calculate?
Answer 5: As your comment, we revised Rf was administered by gavage (line 102 in the revised manuscript)
.
Comment 6: Authors should add the molecular weight next to their blots. It can simply be noted to the side of the blot in the figures.
Answer 6: As your comment, the information about molecular weight was inserted next to each blot.
Comment 7: I want to commend the authors for the blots in figure 5. Should sample lanes in quadruplicate strengthens your findings and shows transparency.
Answer 7: I appreciate your comment.
Comment 8: Authors need to list the limitations of the study. For example, the translation of this to humans is unknown. Also, the most effective dosing strategies remain to be discovered so these findings may not translate to all dosages and regimens.
Answer 8: I agree with your comment. As your comment, the list of limitations was described in lines 353-356 of the discussion section (lines 371-374 in the revised manuscript).

Reviewer 2 Report
The study entitled “Ginsenoside Rf enhances exercise endurance by stimulating myoblast differentiation and mitochondrial biogenesis” from Korea is interesting. In a mouse model, authors demonstrated the influential role of Rf pretreatment in myoblast differentiation that promotes exercise performance. The findings are reasonable and data supported authors’ hypothesis. Although this manuscript is merit enough to publish, there are some issues which should be addressed by the authors.
Comments:
1. Title is good, but experimental condition/model is missing.
2. Abstract: Line 20-the treatment details of G-Rf are missing.
3. Authors provided more details on methods in the Abstract. Instead, the detailed results should be presented to strengthen the paper.
4. In vitro studies: The concentrations of Rf need to be provided.
5. Background: Authors well explained the mechanism and key signaling molecules involved in myogenesis and mitochondrial biogenesis. But the role of Rf and rationale with previous studies need to be explained.
6. The importance or necessity of this study also needs to be included.
7. Please include total number of mice and average weights.
8. Authors should provide more details on animal grouping, treatment and sample collection.
9. Western blot: Please describe the protocol that exactly used in this study with details of primary antibodies and respective secondary antibodies.
10. Biochemical analyses and/or exercise performance from animal studies are completely missing.
11. Rf dosage: Please provide the supporting reference to state the high low dose of Rf to mice.
12. It’s curious…. why authors didn’t estimate any signaling molecule in the muscles of mice?
13. Authors provided good enough evidence from in vitro studies, but not from in vivo studies.
Author Response
Comments and Suggestions for Authors
The study entitled “Ginsenoside Rf enhances exercise endurance by stimulating myoblast differentiation and mitochondrial biogenesis” from Korea is interesting. In a mouse model, authors demonstrated the influential role of Rf pretreatment in myoblast differentiation that promotes exercise performance. The findings are reasonable and data supported authors’ hypothesis. Although this manuscript is merit enough to publish, there are some issues which should be addressed by the authors.
Comments 1: Title is good, but experimental condition/model is missing.
Answer 1: I appreciate your comment. As your comment, the title was changed to “Ginsenoside Rf enhances exercise endurance by stmulating myoblast differentiation and mitochondrial biogenesis in C2C12 myotubes and ICR mice”.
Comment 2: Abstract: Line 20-the treatment details of G-Rf are missing.
Answer 2: As your comment, “G-Rf increased the swimming duration of the experimental animal.” was changed to “The swimming duration of experimental animals was increased by oral gavage administration of G-Rf.” (lines 21-22 in the revised manuscript)..
Comment 3: Authors provided more details on methods in the Abstract. Instead, the detailed results should be presented to strengthen the paper.
Answer 3: As your comment, the Abstract has been revised to describe the results (lines 21-25 in the revised manuscript).
Comment 4: In vitro studies: The concentrations of Rf need to be provided.
Answer 4: As your comment, the information on G-Rf concentration was inserted in section 2.4 (line 117 in the revised manuscript).
Comment 5: Background: Authors well explained the mechanism and key signaling molecules involved in myogenesis and mitochondrial biogenesis. But the role of Rf and rationale with previous studies need to be explained.
Answer 5: As your comment, the correlation between the role of G-Rf and previous studies was described in the discussion section (lines 367-369 in the revised manuscript).
Comment 6: The importance or necessity of this study also needs to be included.
Answer 6: In the introduction section (lines 75-81 in the revised manuscript) and discussion section (lines 365-369), the necessity of this study was inserted.
Comment 7: Please include total number of mice and average weights.
Answer 7: As your comment, an explanation about total number of mice and average body weight was inserted in lines 100, 191, and 199-200.
Comment 8: Authors should provide more details on animal grouping, treatment and sample collection.
Answer 8: As your comment, an explanation about animal grouping, treatment, and sample collection was described as shown in section 2.2 and 2.3.
Comment 9: Western blot: Please describe the protocol that exactly used in this study with details of primary antibodies and respective secondary antibodies.
Answer 9: Section 2.1 describes the primary and secondary antibodies used in this study.
Comment 10: Biochemical analyses and/or exercise performance from animal studies are completely missing.
Answer 10: Forced swimming test and ATP biochemical analysis were performed in this study, and the methods and results have been described in sections 2.3, 2.5, 3.1, and 3.2.
Comment 11: Rf dosage: Please provide the supporting reference to state the high low dose of Rf to mice.
Answer 11: In the discussion section (lines 299-304), the rationale for determining the G-Rf dosage is presented.
Comment 12: It’s curious…. why authors didn’t estimate any signaling molecule in the muscles of mice?
Answer 12: The protein expression of AMPK signaling-related molecules was measured in mice tibialis anterior muscle tissue. These data were shown in Figure 5C.
Comment 13: Authors provided good enough evidence from in vitro studies, but not from in vivo studies.
Answer 13: Figures 1 and 5C indicate that G-Rf administration enhances exercise endurance and mitochondrial biogenesis through AMPK, Akt, p38 signaling pathways in the mice model.
